# Unlocking the Therapeutic Potential of Marine Collagen: A Scientific Exploration for Delaying Skin Aging

**DOI:** 10.3390/md22040159

**Published:** 2024-03-30

**Authors:** Azizur Rahman, Rameesha Rehmani, Diana Gabby Pirvu, Siqi Maggie Huang, Simron Puri, Mateo Arcos

**Affiliations:** 1Centre for Climate Change Research (CCCR), University of Toronto, ONRamp at UTE, Toronto, ON M5G 1L5, Canada; rameesha.rehmani@mail.utoronto.ca (R.R.); gabrielapirvu@hotmail.com (D.G.P.); sqmaggie.huang@mail.utoronto.ca (S.M.H.); simron@climatechangeresearch.ca (S.P.); mateo.arcos@mail.utoronto.ca (M.A.); 2A.R. Environmental Solutions, ICUBE-University of Toronto, Mississauga, ON L5L 1C6, Canada; 3AR Biotech Canada, Toronto, ON M2H 3P8, Canada; 4Department of Biological Anthropology, University of Toronto, Mississauga, ON L5L 1C6, Canada; 5Department of Ecology and Evolutionary Biology, University of Toronto, St. George, Toronto, ON M5S 3B2, Canada; 6Computer Science, Mathematics and Statistics, University of Toronto, Mississauga, ON L5L 1C6, Canada

**Keywords:** marine collagen, biopeptide, antioxidant, skin, anti-aging, prevention, bone regeneration, extracellular matrix (ECM), fish collagen

## Abstract

Aging is closely associated with collagen degradation, impacting the structure and strength of the muscles, joints, bones, and skin. The continuous aging of the skin is a natural process that is influenced by extrinsic factors such as UV exposure, dietary patterns, smoking habits, and cosmetic supplements. Supplements that contain collagen can act as remedies that help restore vitality and youth to the skin, helping combat aging. Notably, collagen supplements enriched with essential amino acids such as proline and glycine, along with marine fish collagen, have become popular for their safety and effectiveness in mitigating the aging process. To compile the relevant literature on the anti-aging applications of marine collagen, a search and analysis of peer-reviewed papers was conducted using PubMed, Cochrane Library, Web of Science, and Embase, covering publications from 1991 to 2024. From in vitro to in vivo experiments, the reviewed studies elucidate the anti-aging benefits of marine collagen, emphasizing its role in combating skin aging by minimizing oxidative stress, photodamage, and the appearance of wrinkles. Various bioactive marine peptides exhibit diverse anti-aging properties, including free radical scavenging, apoptosis inhibition, lifespan extension in various organisms, and protective effects in aging humans. Furthermore, the topical application of hyaluronic acid is discussed as a mechanism to increase collagen production and skin moisture, contributing to the anti-aging effects of collagen supplementation. The integration of bio-tissue engineering in marine collagen applications is also explored, highlighting its proven utility in skin healing and bone regeneration applications. However, limitations to the scope of its application exist. Thus, by delving into these nuanced considerations, this review contributes to a comprehensive understanding of the potential and challenges associated with marine collagen in the realm of anti-aging applications.

## 1. Introduction

Collagen is a fibrous protein that provides support to various structures of the body, such as the skin, cartilage, and bones [1,2,3]. Functioning as a crucial structural and connective component of the extracellular matrix (ECM), collagen helps regulate cell growth, adhesion, and migration [4,5]. As a naturally abundant protein found in all animals, there are 28 different types of collagens, which account for approximately 30% of the total protein found in the body [2]. Type 1 collagen is the most abundant type and provides support to most tissues of the body, such as the skin and muscles. Type II is responsible for the maintenance and repair of cartilage [2]. Type III is the main element of tissue sealants and reticular fibers commonly found in blood vessels and muscles [2]. Type IV is a key element in the basement membrane, functioning as a barrier between tissues, and can act as a diabetic neuropathy indicator [2]. Finally, Type V is the main collagen in corneal solutions and is found in the placenta and hair [2].

Collagen is often used as a regenerative biomaterial due to its high biodegradability, solubility, and tensile strength [4,5,6,7,8,9]. Its low immunogenicity and excellent biocompatibility have prompted extensive research into its application as a polymer across various biomedical products, including cosmetics and pharmaceuticals [5]. Moreover, collagen serves as a safe and efficient biomaterial in tissue engineering and clinical settings [9,10]. The food industry also exhibits a substantial demand for collagen due to its elevated protein content and beneficial functional attributes, including water absorption capacity and emulsion-forming ability [5]. However, the natural degradation of collagen accelerates with age, which can impact skin elasticity, wound healing, bone density, and even immune and neural function [4,11,12,13]. Skin aging results from diminished collagen density and dermal thickness, alongside reduced synthesis and replacement of crucial structural proteins [5]. The effects of reduced collagen density especially impact the dermis layer of the skin, resulting in notable signs of aging such as increased wrinkling, sagging, laxity, and a textured appearance [14,15]. 

While the aging process of the skin is considered complex, the incorporation of marine collagen in anti-aging supplements has been used to treat select skin concerns, including visible signs of aging [1]. In particular, the application of marine collagen peptides (MCPs) emerges as a promising therapeutic according to multiple animal and in vitro studies [7,11,16,17,18]. MCPs are obtained by hydrolyzing collagen into small peptides of low molecular weight to improve bioavailability and absorption [19]. MCPs can exert bioactive properties, including anti-microbial and antioxidant functions [19]. Thus, MCPs are commonly utilized in cosmeceutical skin products for their anti-aging properties [19,20,21,22]. Over the past decade, there has been a remarkable surge in market demand for marine-based cosmetics [21].

While the previous literature has demonstrated the effectiveness of bioactive natural peptides in mitigating the effects of aging across diverse models, including cell studies, animal studies, and clinical trials [1,5,23,24,25], there is limited information regarding the diversity of anti-aging collagen peptides found in marine organisms. Various bioactive compounds can be sustainably extracted from marine waste and harnessed as potent ingredients for the formulation of cosmetic products, reducing environmental pollution and lowering production costs [25,26,27,28,29]. Examples include collagen derived from fish waste and chitin obtained from crustacean waste, which can be incorporated into cosmetic formulations targeting anti-wrinkle and skin barrier enhancements [28]. While fish are widely used as food resources, there is limited utilization of marine proteins from other species such as sea cucumbers, sea urchins, mussels, and various kinds of algae. Thus, this review addresses this gap by presenting recent insights into the anti-aging potential of bioactive collagen peptides sourced from under-utilized marine resources, examining examples that can scavenge free radicals in vitro and showcase clinical benefits for the skin and body [15,28]. Interestingly, the potential combination of CRISPR technology with marine collagen offers a novel perspective for groundbreaking anti-aging treatments. This synergy harnesses CRISPR’s precision in gene editing to specifically target aging-related genes, complemented by the supportive properties of marine collagen. The result is an innovative approach with enhanced therapeutic effects, particularly in skin elasticity and hydration [30]. The findings lay the groundwork for the development of revolutionary anti-aging collagen treatments derived from underexplored marine organisms. 

Table 1 displays the five common types of collagens along with their functions. Marine collagen is predominantly Type I collagen, which is the primary component of the calcium-depleted tissue of the teeth and bone [2]. It is found in the skin, in tendons, in the vasculature of the lungs, and in the heart [2]. Table 1 significantly highlights the use of porcine collagen (Type I and Type III collagen), which is essential in the prevention and treatment of osteoporosis [2].

Our literature review explores the anti-aging activities of collagen peptides from marine organisms, focusing on their capacity to regulate oxidative stress in diverse models including cells, fruit flies, nematodes, mice, and humans [1]. By analyzing the findings of these papers, we aim to contribute valuable insights that help enhance the utilization of marine sources for anti-aging applications. However, limitations regarding the lack of long-term studies may hinder its potential use. Thus, this review highlights the applications and limitations of anti-aging marine collagen research while outlining future directions for this field. 

## 2. Marine Collagen: Effects on Skin Aging 

The anti-aging industry is growing rapidly with the release of new supplements and nutraceuticals that promise youthful skin, better joint and bone health, and even stronger immune systems [29,31,32]. Among the most popular products is marine-derived collagen used for skin health and restoration. Several papers have cited the effects of collagen supplementation on skin appearance [7,8,20,21,22,32]. 

A recent study by Lee et al (2022) investigated the importance of collagen formulation on anti-aging efficacy [32]. Only compounds with low molecular weights may penetrate the skin barrier, limiting the efficacy of intact collagen application, and oral administration of collagen peptides is limited by their poor stability and absorption in the gastrointestinal tract [32]. Thus, to increase absorptive ability, the fish collagen was hydrolyzed into small, bioactive collagen peptides and administered as an orally disintegrating film, allowing the collagen to be directly absorbed into the bloodstream [32]. 

After applying the film for 12 weeks, the authors concluded that fish-derived collagen administered as an orally disintegrating film was effective at significantly reducing skin wrinkle depth and number, as well as increasing skin elasticity and density in women aged 20 to 60 years old [32]. As evident in Figure 1, at individual ages, a number of changes occur in the density and structure of collagen fibers [33]. Figure 1 displays a decline in the number of collagen fibrils and the size of the fibroblast cells as the skin ages, emphasizing the importance of collagen fibrils in the maintenance of cell size and skin elasticity for healthy skin [33]. 

Further on, the results of a 2018 randomized placebo-controlled trial revealed the hydrating, anti-aging effects of a low molecular weight collagen hydrolysate obtained from sutchi catfish skin [7]. After 6 weeks, skin hydration was 7.23-fold higher in the treatment group compared to the placebo (*p* < 0.001). This hydrating benefit was observed after 12 weeks as well, although at only 2.9-fold higher than the placebo (*p* < 0.01) [7]. Moreover, wrinkle formation was also reduced, considering parameters such as skin roughness, smoothness depth, and visual grading, demonstrating the anti-aging potential of hydrolyzed marine collagen on the skin of older adults [14]. Longer-term studies should be conducted to determine whether this beneficial effect holds true over time. 

Another study used a mouse model of aging to demonstrate that marine collagen may restore a youthful skin appearance [16]. In this study, mice were fed a collagen hydrolysate-containing diet derived from fish scales for 12 weeks. Notably, the epidermal barrier and dermal elasticity dysfunctions observed in the aging group were significantly attenuated in the collagen hydrolysate treatment group after 2 weeks [16]. Further on, these positive effects were maintained for the entirety of the study duration, demonstrating a prolonged restoration of skin elasticity and water content following collagen supplementation [16].

## 3. Marine Bioactive Peptides: Antioxidant and Anti-Carcinogenic Roles

On top of providing valuable sources of nitrogen and amino acids, many bioactive MCPs have demonstrated powerful antioxidant, anti-microbial, and immunomodulatory effects. The active peptide products isolated from fish, sea cucumbers, sponges, urchins, mussels, and other marine life have shown the potential to lower oxidative stress, inhibit cellular senescence, and extend lifespans in multiple animal studies of aging [17,34,35]. For example, one study found that jellyfish collagen hydrolysate (JCH) improved the exercise tolerance of mice in a dose-dependent manner after 6 weeks [17]. In the same study, the authors used d-galactose to induce the aging process in mice, then investigated the effect of JCH on oxidative stress by measuring malondialdehyde (MDA), superoxide dismutase (SOD), and glutathione peroxidase (GSH-Px) activity [17]. MDA is a product of lipid peroxidation, which increases with age, whereas SOD and GSH-Px are enzymatic antioxidant systems that neutralize free radicals implicated in the aging process, wherein activity declines with age [18]. Significantly, the 6-week administration of JHC inhibited the decrease in GSH-Px/SOD activity and the increase in MDA seen in the ageing model [17]. These results are displayed in Figure 2 [17], showcasing the powerful in vivo antioxidant capacity of marine peptides and demonstrating their benefit in anti-aging products. 

Similarly, Liang et al (2010) discovered that rats fed chum salmon MCPs over the course of their lifespan showed increased GSH-Px and SOD enzymatic activity compared to control rats; however, this change was only significant in rats older than 12 months [18]. Further on, the observed age-related MDA increase was attenuated in MCP-treated rats, suggesting that MCPs can interfere with the cellular and physiological effects of aging by exerting antioxidant effects [18]. Significantly, this study also demonstrated that MCP-treated rats on average had longer life spans and better survival outcomes after 28 months [18]. MCP treatment also delayed tumor growth, decreased tumor size and number, and lowered the incidence of deaths from tumors after 16 months when compared to the control [18]. Genetic mutations increase in frequency with age, predisposing the cells to various oncogenic processes; thus, MCP may act in a protective, anti-carcinogenic capacity to slow the progression of aging-related diseases such as cancer [18]. Taken together, these findings suggest that marine collagen may exert antioxidant capabilities that interfere with the aging process, leading to longer, healthier lives. 

## 4. Marine Collagen in Tissue Engineering for Anti-Aging 

Marine collagen is recognized for its bioactive properties and is used in skin tissue engineering due to its water solubility, metabolic compatibility, and accessibility. It has shown effectiveness in healing skin injuries of varying severity and in delaying aging processes, promoting keratinocyte and fibroblast migration, and vascularization of the skin [36]. In animal model studies, marine collagen from different species has shown promising results in skin tissue healing [37,38]. Treatments using marine collagen have led to increased deposition of granulation tissue, enhanced re-epithelialization, stimulated neoangiogenesis, and improved the morphological aspect of wounds [38]. These findings underscore its potential in tissue engineering and wound healing applications [37,38]. The marine environment has been a significant source of biological macromolecules for developing tissue-engineered substitutes with wound-healing properties. These molecules play a key role in enhancing the wound-healing process and are crucial in advancing wound-care management [34]. Moreover, studies on collagen-derived peptides from the marine sponge *Chondrosia reniformis* reveal their antioxidant activity, ability to stimulate cell growth, and protection against UV-induced cell death [34]. These peptides have shown no toxicity in cell lines, and their significant ROS scavenging activity indicates their potential in drug and cosmetic formulations, especially for damaged or photoaged skin repair [34]. 

Marine collagen’s role in bioprinting and scaffold development is pivotal for tissue regeneration. Marine collagen, particularly Type I, is ideal for creating 3D bioprinted structures due to its biocompatibility, biodegradability, and low immunogenicity [39,40]. These structures mimic the natural ECM of human tissues, which is essential for effective tissue regeneration. In tissue engineering scaffolds, marine collagen offers a structure that supports cell attachment, proliferation, and differentiation, key factors for tissue repair and regeneration [40]. These properties of marine collagen are particularly significant in anti-aging applications, as they support the growth and repair of various tissues, including skin, bone, and cartilage. As aging is associated with the degradation of these tissues, marine collagen scaffolds can be used to replace or support damaged tissues, thereby counteracting some of the effects of aging. Ongoing research is optimizing marine collagen properties for bioprinting and scaffold design, enhancing its mechanical strength, stability, and compatibility with human tissues. The development of hybrid scaffolds, combining marine collagen with other biomaterials, is also an area of interest to improve functionality and efficacy in tissue regeneration [4,40,41,42].

## 5. The Integration of CRISPR Technology with Marine Collagen

The integration of CRISPR technology with marine collagen is an innovative area of research, combining the genetic editing capabilities of CRISPR with the beneficial properties of marine-derived collagen. Marine collagen has shown promise as a biomaterial in various applications, including wound healing, skin anti-aging, and bone regeneration. Its biocompatibility makes it an excellent candidate for tissue engineering and regenerative medicine. On the other hand, CRISPR technology offers a groundbreaking approach to gene editing, allowing for precise modifications at the DNA level. An overview of this technology can be seen in Figure 3. CRISPR technology has been making strides in various medical applications, including the development of more refined editing techniques such as base and prime editing. These newer methods aim for uniform and predictable gene-editing results while minimizing potential risks associated with traditional CRISPR-Cas9 techniques, such as the creation of double-strand DNA breaks [43,44,45]. 

Dermatological applications of CRISPR technology have been highly promising [30,47,48]. Despite there being no direct research on using CRISPR technology on marine collagen, the integration of these two fields could potentially lead to effective anti-aging treatments. For instance, CRISPR technology could be employed to target and modify specific genes associated with aging and skin degeneration. The progressive alterations observed in aging skin are now being comprehensively observed at both the molecular and cellular levels, leading to enhanced insights into the structural and functional decline resulting from these changes [33]. By precisely editing these genes, it might be possible to slow down or reverse certain aging processes at a molecular level. Meanwhile, marine collagen could play a supportive role in this integration. Its ability to enhance cell viability and support tissue regeneration could be crucial in facilitating the effectiveness of CRISPR-mediated gene edits. For example, in a scenario where CRISPR is used to edit genes related to skin elasticity, marine collagen could provide the necessary ECM support, enhancing the overall therapeutic effect.

Another potential approach to integrating CRISPR with marine collagen could focus on enhancing skin hydration and barrier function. This would involve using CRISPR to edit genes crucial for maintaining skin moisture, such as those involved in hyaluronic acid synthesis. Concurrently, marine collagen could be developed as a topical delivery system for CRISPR components, leveraging its skin absorption properties and biocompatibility. This could involve encapsulating CRISPR components (such as Cas9 and guide RNA) within marine collagen-based nano-carriers that can penetrate the skin layers. 

While direct research on the integration of CRISPR technology with marine collagen in anti-aging has yet to be performed, the combination of CRISPR’s precision in genetic editing and marine collagen’s supportive properties presents a possibility. Future research in this area could lead to innovative and effective anti-aging therapies, potentially revolutionizing the way we approach aging and skin health.

## 6. Marine Collagen Use: The Pros and Cons

The ECM plays a fundamental role in ensuring cell integrity and aiding in various cell functions, such as proliferation, differentiation, migration, and adhesion [41]. Marine organisms, including fish, jellyfish, sponges, and other invertebrates, provide a valuable source of collagen that is free from religious restrictions and animal pathogens (Figure 4). This type of collagen is metabolically compatible and has advantages over other sources [41]. Fish skin is often used to extract Type I collagen because it is abundant and not suitable for industrial use. Overall, marine sources of collagen are a safe, convenient, and promising option. The combination of biomaterials and single gene delivery has shown promising potential for tissue engineering. Studies have found that marine collagen from organisms such as fish, jellyfish, and sponges can promote wound healing, enhance blood circulation, and prevent infection [41]. Additionally, marine collagen has anti-aging properties that have been demonstrated in mice with osteoporosis [41]. It can increase bone mineral density, protect against bone loss and osteoarthritis, induce plastic differentiation, and even improve skin elasticity while slowing the aging process [41,49]. Finally, drug delivery and immobilization are two ways marine collagen is used within the human body [41]. Marine collagen offers several advantages compared to other popular sources of collagen, notably bovine or porcine collagen. One significant benefit is its resource abundance, as marine collagen is derived from the massive amounts of marine waste produced by the fishing industry, helping reduce environmental contamination while providing high yields at lower costs [50]. Marine collagen also presents with a higher biocompatibility and no disease transmission risk; thus, considering mammalian collagen has been associated with incidents of prion transmission leading to conditions such as bovine spongiform encephalopathy (BSE), marine collagen is considered a safer alternative [51]. However, marine collagen sources, such as fish and marine sponges, still carry the threat of allergens [52,53]. Allergenicity refers to the likelihood of a product causing an adverse immune response in the body. Depending on the type of fish or fish product that the collagen is sourced from, the level of allergenicity will vary. For example, collagen from bony fish has been shown to have higher allergenicity than collagen from cartilaginous fish [53]. To reduce the likelihood of adverse effects, standardized methods for extraction and purification of marine collagen need to be further investigated. Ultimately, the anti-aging effect of marine collagen can be evident throughout the body both externally and internally. Externally, through reversing the aging effects of the skin, and internally, through regulating bone health, tissue regeneration, and dietary and metabolic processes [54,55]. Together, these effects improve overall health, skin appearance, and well-being.

### 6.1. Hydration

Hyaluronic acid plays an important role in skin moisture retention [56]. Previous literature on oral collagen supplements has shown evidence of targeting age-related concerns and improving skin integrity [57,58]. Marine collagen and collagen peptides, especially from fish, have demonstrated significant effects on skin hydration. When administered orally, collagen hydrolysates can restore the production of hyaluronic acid to improve skin hydration [57,58]. Other studies have reported that canary seed peptides (CSPs) show promising results for skin aging treatments [59]. However, fish collagen is considered an optimal source due to its diverse amino acid compositions and high bioavailability [60]. Limited research on chicken-derived collagen suggests potential benefits, but more studies are needed for conclusive results [57].

### 6.2. Elasticity

Elastin and microfibrils in the elastic fabric network relay elasticity and resilience to the skin. Consuming oral collagen has been expressed to improve skin elasticity, resulting in increased levels of Type I collagen [56]. Numerous studies exhibit positive effects on skin elasticity, including improvements in surface elasticity [56]. Collagen peptides have been found to increase collagen content and improve skin laxity in a variety of animal and human studies. However, there are limitations to this research, such as differences in the duration and dosage quantities, small sample sizes, and self-reported skin elasticity measurements [61].

## 7. Collagen as a Biomaterial for Tissue Engineering

One of the most common and prominent biomaterials in tissue engineering and regenerative medicine is collagen, such as collagen proteins in the ECM of marine invertebrates [4] (Figure 5). Although fish collagen peptides (FCPs) have been used as a dietary supplement, little is known about how they affect cellular function in the human body [62]. ECM replacements can significantly affect cell proliferation and function based on recent research [36,63]. These extracellular matrixes, however, are mainly used in a general sense and are not yet tailored to certain cell types [63]. This paper focuses on ECM-based coating substrates tailored to the individual needs of skin, skeletal muscle, and liver cell cultures [63]. With ongoing advancements, neural tissue engineering (NTE) indicates significant potential to treat a number of debilitating neurological illnesses [5]. For NET design strategies that facilitate axonal growth and neural and non-neural cell differentiation, choosing the best scaffolding material is essential. As the nervous system is naturally resistant to regeneration, collagen is often used in NTE applications [5]. It can function with neurotrophic factors, neural growth inhibitors, and other compounds that promote neural growth [5]. It can also be used for neural repair and thus mitigate neurodegenerative diseases that come with age [5]. The ECM is a powerful structure that influences the cells in contact with it [64]. A poor prognosis has been linked to the composition and collagen density of the tumor-specific ECM in a number of cancer forms [64]. The cause of this correlation is still mainly a mystery [64]. Collagen can stimulate the development and migration of cancer cells, although collagen has been found in recent research to influence the activity and phenotype of T cells and tumor-associated macrophages (TAMs), two types of immune cells that infiltrate tumors [64]. 

## 8. Applications of Marine Collagen in the Cosmetic Market

The cosmeceutical industry is flooded with a variety of anti-aging products that claim to address wrinkles, fine lines, and other signs of aging through various mechanisms of action. Some of the most popular products on the market include retinoids such as retinol, retinyl esters, and retinaldehyde—vitamin A derivatives known for their ability to stimulate collagen production and promote cell turnover, thereby increasing skin elasticity and reducing the appearance of fine lines and wrinkles [52]. Alternatively, vitamin C- and vitamin E-based serums provide anti-aging benefits through their powerful antioxidant effects that protect the skin against UV-induced photodamage and oxidative stress [52]. Vitamin C stimulates collagen synthesis, and both vitamins have anti-inflammatory functions that aid in wound healing. Another widely used anti-aging ingredient is hyaluronic acid, a hydrating glycosaminoglycan (GAG) that can act as a barrier against trans-epidermal water loss to retain skin moisture and reduce the appearance of fine lines [52]. Other popular products include alpha hydroxy acids (i.e., lactic acid, glycolic acid, citric acid) and beta hydroxy acids (i.e., salicylic acid) that exfoliate the skin, promote cell turnover, and facilitate GAG and collagen synthesis to improve skin texture and tone [50,52]. Finally, numerous bioactive peptide formulations can help stimulate collagen production and improve skin elasticity [52,54]. For example, extracts from brown algae have proven a plentiful resource for anti-inflammatory and antioxidant compounds (Figure 6) [66]. Because of their photoprotective properties, these bioactive peptides can be used in cosmetic preparations for anti-aging skincare and sunscreen [66]. In one study, the brown algae, *Ericaria amentacea*, showed dose-dependent in vitro activity for reducing various markers of oxidative stress, inflammation, and collagen and hyaluronic degradation. The results of various antioxidant assays are displayed in Figure 7 [66], demonstrating the remarkable potential of anti-aging cosmetics.

While these products demonstrate numerous anti-aging benefits, there are associated drawbacks that may limit their efficacy and applicability. For one, prescription-strength retinoid formulations and AHAs may induce adverse reactions such as skin irritation, burning, and dermatitis [52]. In addition, the oxidation of retinol, vitamin C, and vitamin E over time poses a problem for the stability of the product, which can affect the overall quality and efficacy of the cosmetic preparation [41]. Further on, in rare cases, topical application of vitamin E has been linked to cases of contact dermatitis, erythema multiforme, and xanthomatous reaction [52]. As a result, recent trends in the anti-aging industry have demonstrated rising consumer interest in the natural bioactive compounds found in marine collagen rather than synthetic ingredients [66,67,68]. A recent study of the Portuguese anti-aging cosmetic market revealed a 27% increase in marine collagen cosmetics from 2011 to 2018, with red algae being the most widely used marine ingredient [66]. 

A potential explanation for the increasing popularity of marine collagen products may lie in the manufacturing, safety, and efficacy advantages that they provide. For one, marine sources are biodiverse, abundant, and easy to cultivate and modulate during their life cycles. Furthermore, marine collagen is easily absorbed by the body and efficiently utilized for collagen synthesis. These characteristics make it possible to harness the production of specific bioactive compounds involved in collagen synthesis and wound healing [50,67]. Further on, collagen-based cosmetics predominantly utilize Type I collagen, valued for its moisturizing, anti-wrinkle, anti-aging, wound-healing, and UV radiation protection properties [20]. Figure 8 further illustrates the various utilizations and applications of marine collagen in the cosmetic market. 

Although marine collagen has a wide range of benefits, the effectiveness of any anti-aging product can vary depending on individual genetic and environmental factors, including chronic autoimmune conditions, exposure to sun and air pollution, and lifestyle choices [20]. 

## 9. Concluding Remarks

This literature review highlights the diverse biomedical anti-aging applications of marine collagen, establishing it as a versatile biomaterial in tissue engineering and regenerative medicine. Marine collagen’s unique properties, such as promoting osteogenesis, collagen synthesis, and anti-inflammation, highlight its pivotal roles in accelerating the healing process, promoting skin health, and maintaining free radical homeostasis. Compared to land animal sources, advantages such as metabolic compatibility, safety, and environmental sustainability further position marine collagen as a compelling choice for anti-aging applications. Utilizing readily available marine waste from the fishing industry ensures cost-effective production and addresses environmental concerns. Marine collagen also proves valuable in vascular tissue engineering, demonstrating promise in crafting advanced scaffolds for vascular grafts, enhancing mechanical strength, and fostering vascular endothelial cell development. Ongoing research and innovation efforts focused on marine collagen extraction, processing, and application, as well as synergy with anti-aging CRISPR technology, underscore its continued importance in advancing the fields of tissue engineering and biomedicine. To discover further anti-aging applications of various marine collagen sources, further research efforts should continue exploring this protein’s remarkable therapeutic efficacy and versatility.

## Figures and Tables

**Figure 1 marinedrugs-22-00159-f001:**
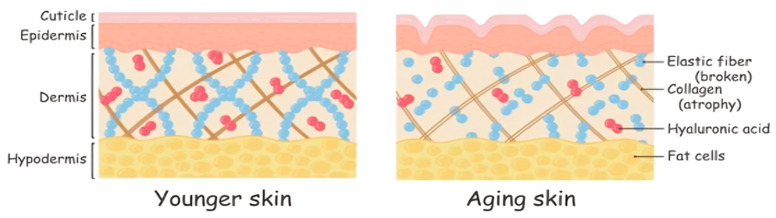
Illustrates the structural differences between younger and aging skin. In young human skin dermis, collagen fibrils are intact and normal in size (**left**) in contrast with reduced collagen fibrils in aged human skin dermis which leads to a reduction in cell size (**right**). The aging skin on the right shows a reduction and fragmentation of collagen fibers, broken elastic fibers, and diminished Hyaluronic Acid (red dots), leading to thinner fat layers and an overall loss of structural integrity and elasticity.

**Figure 2 marinedrugs-22-00159-f002:**
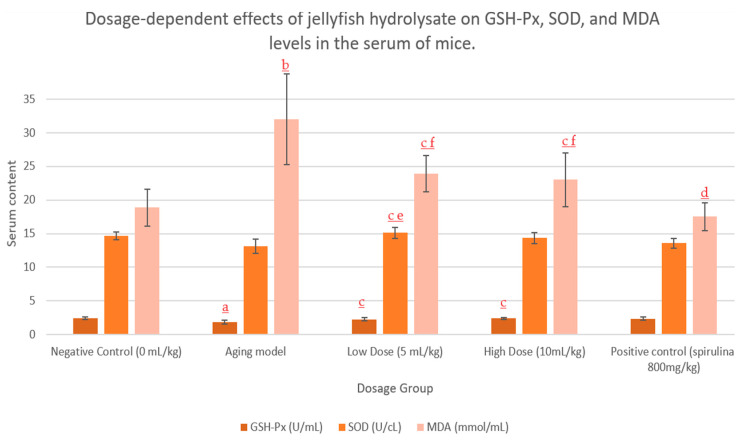
Serum levels of GSH-Px, SOD, and MDA in mice. Data from J.F. Ding et al. (2011). a = *p* < 0.05, b = *p* < 0.01 compared to control. c = *p* < 0.05, d = *p* < 0.01 compared to aging model. e = *p* < 0.05, f = *p* < 0.01 compared to positive.

**Figure 3 marinedrugs-22-00159-f003:**
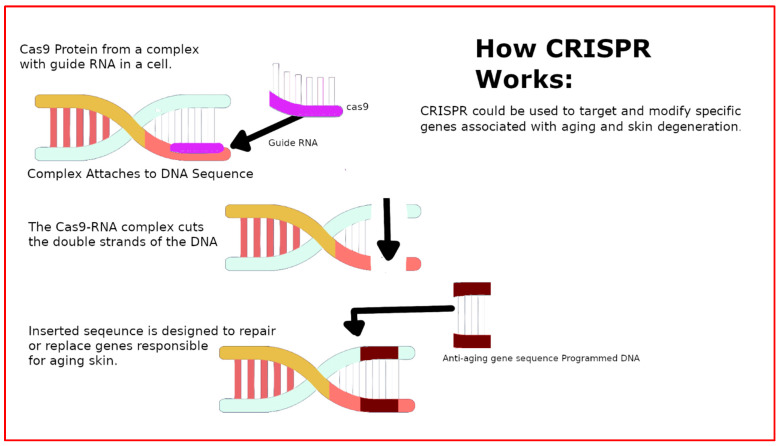
Illustration of the CRISPR-Cas9 Mechanism for Skin Regeneration. This graphic outlines the use of CRISPR-Cas9 technology for targeted gene editing in eukaryotic cells, specifically for skin regeneration. The process begins with the Cas9 protein forming a complex with a guide RNA that is complementary to a specific gene sequence associated with skin aging. This complex then locates and binds to the target DNA sequence, where Cas9 makes a precise cut. A new DNA sequence with the desired genetic information can then be inserted at the cut site for potential therapeutic purposes, such as reversing aging effects or repairing skin damage. This advanced molecular technique is also being applied to edit the genetic code of various organisms, encompassing eukaryotic cells similar to those in humans. Specifically, in the context of combating skin aging, this method allows for precise alterations to DNA sequences, facilitating the repair or reversal of age-related genetic changes in the skin. It might also offer a tool for curing genetically based diseases [46].

**Figure 4 marinedrugs-22-00159-f004:**
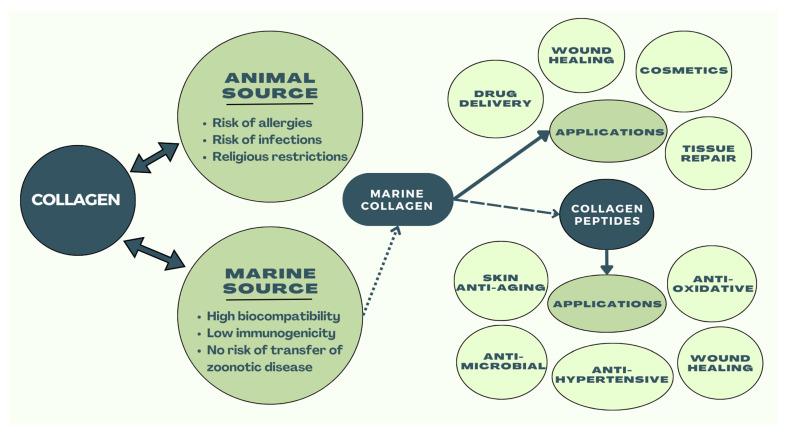
Illustrates biomedical applications and advantages of marine collagen compared to land animal-derived collagen.

**Figure 5 marinedrugs-22-00159-f005:**
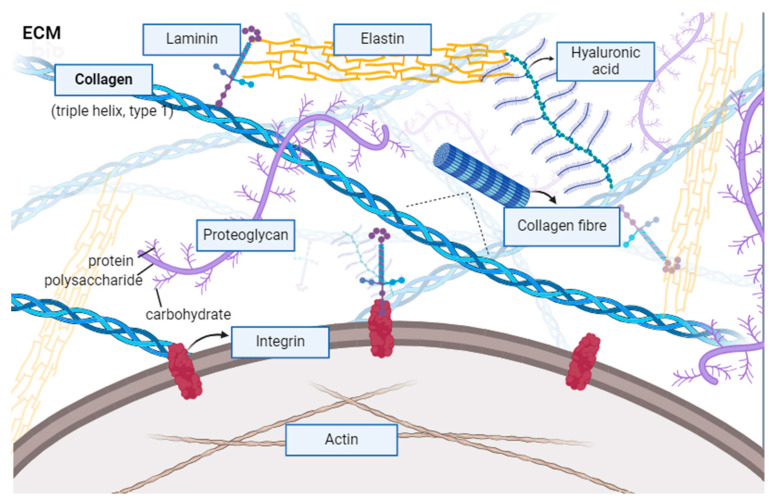
Collagen found in the ECM. The ECM is a dynamic network of proteins and molecules that play a fundamental role in organizing and maintaining tissue structure and function. Components of the ECM include fibrillar proteins (i.e., collagen, elastin) which confer tensile strength and elasticity, adhesive glycoproteins (i.e., fibronectins, laminins) which mediate cell–ECM interactions critical for tissue organization and homeostasis, and proteoglycans (i.e., fibromodulin), which can have biologically active properties (i.e., growth factors) and mediate ECM assembly and organization [65]. Created in Biorender.com.

**Figure 6 marinedrugs-22-00159-f006:**
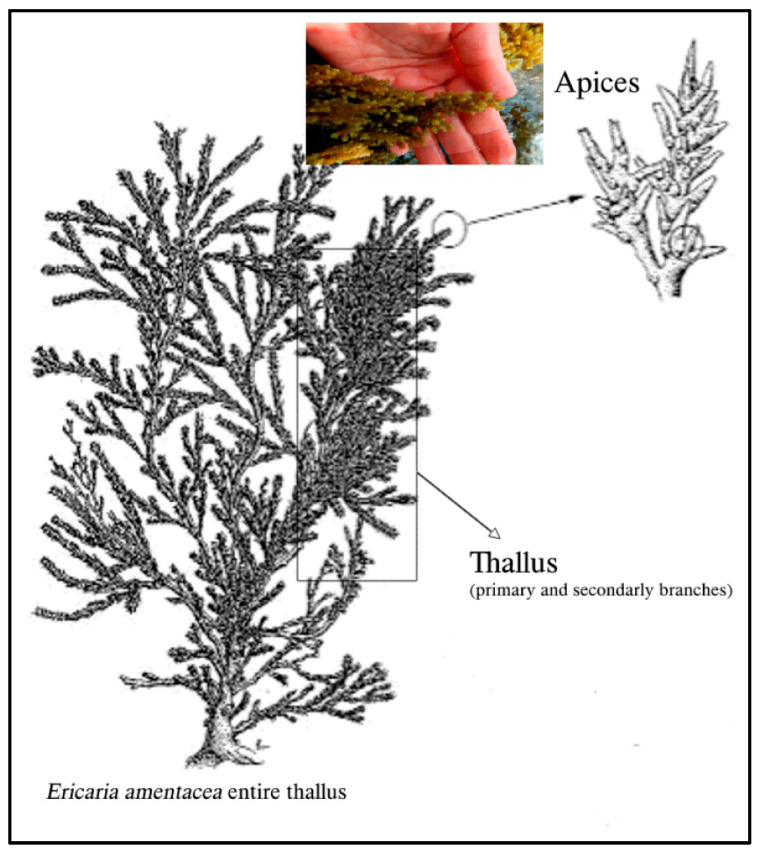
Schematic representation of *E. amentacea* seaweed body parts, and antioxidant activity of *E. amentacea* extracts in spectrophotometric tests.

**Figure 7 marinedrugs-22-00159-f007:**
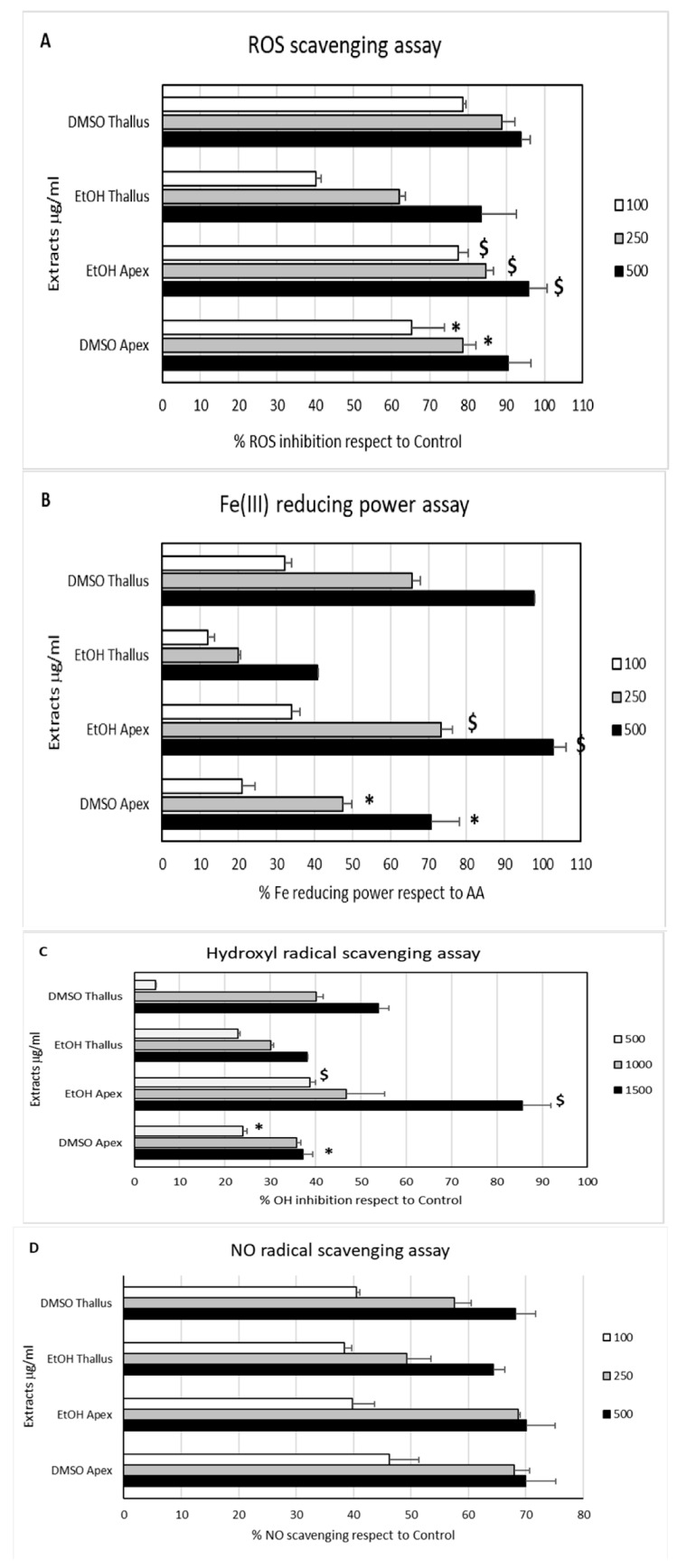
Marine seaweed displays anti-aging properties. In (**A**,**C**), * *p* < 0.05; Tukey of EtOH apex vs. EtOH thallus’ respective concentrations, $ *p* < 0.05. In (**B**), * *p* < 0.05; Tukey of EtOH apex vs. EtOH thallus’ respective concentrations, $ *p* < 0.005. (**A**) ROS scavenging activity. (**B**) Fe (III)-reducing power assay compared to ascorbic acid (AA). (**C**) OH radical scavenging activity. (**D**) NO radical scavenging activity. Taken from Mirata et al. (2023) [66].

**Figure 8 marinedrugs-22-00159-f008:**
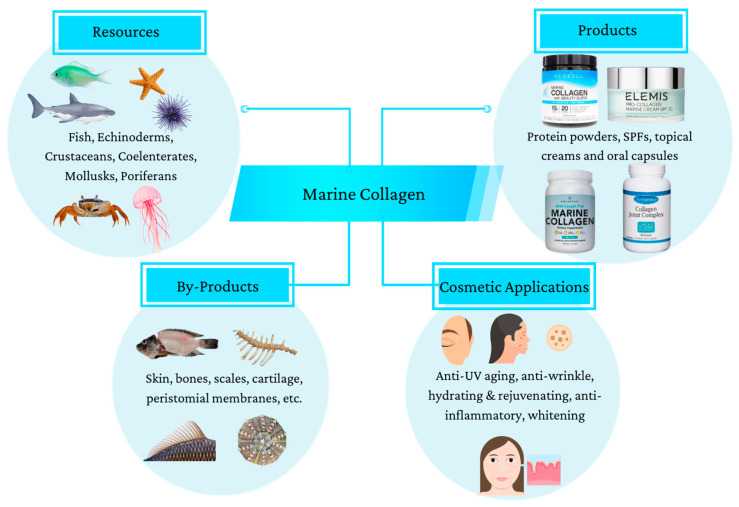
Potential utilization of marine by-products in anti-aging cosmetic preparations.

**Table 1 marinedrugs-22-00159-t001:** The function of the 5 most common types of collagens [2].

Collagen	Function or Application	Tissue or Organ
Type I	the organic part of the bone, membranes for guided tissue regeneration	skin, bone, teeth, tendon, ligament, vascular ligature
Type II	the main constituent of cartilage, cartilage repair, and arthritis treatment	cartilage
Type III	the main constituent of reticular fibers, hemostats, and tissue sealants	muscle, blood vessels
Type IV	the major component of the basement membrane, attachment enhancer of cell culture, and diabetic nephropathy indicator	basal lamina, the epithelium-secreted layer of the basement membrane
Type V	feedstock for biomaterials in corneal treatments	hair, cell surfaces, and placenta.

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
