# Peer review of "Unlocking the Therapeutic Potential of Marine Collagen: A Scientific Exploration for Delaying Skin Aging"

_marinedrugs, 2024, doi:10.3390/md22040159_

Round 1

Reviewer 1 Report

Comments and Suggestions for Authors

I had high expectations from this manuscript which is why I accepted to review it, since the topic of the review is very interesting and of current importance. However, the review is poorly organized, very repetitive, not scientific and overall, of poor quality for a review. It appears that the review was written by authors very naive in this field and in writing reviews. Evidence is that fact that there are only 37 papers cited which is a very low number. Secondly, the inexperience of the authors is also evident from the relevance of the literature cited which in many sections is inappropriate. Just as example, in section 7 on Collagen peptides, line 323, authors refer to an experiment of a study, but they cite a review (ref 20). The same occurs again in section 6.1, line 294 where they mention chicken-derived collagen, but they cite a review (ref. 29). Again on line 173, they report on a study performed on mice, but they cite a review (ref. 1). These are just a few examples, of improprer use of citations and references. The review has many repetitions in context and seems rather jumbled in parts. What is more important, none of the figures and tables reported are of the authors own making, but all taken and copied from other papers. The legends are in many cases inappropriate.

Figure 1. This figure does not display normal collagen fibrils that are obtained from marine sources. It just displays how collagen and fibroblasts change from young to aged skin. The figure is really over-simplistic for a review and should be redone with proper legend description. Furthermore, the legend reports that aged human skin is observed in vivo, which is absolutely not true. This is not shown.

Figure 4. Very poor quality, blurred and unclear. What type of cells are shown in the left panel is not reported in the legend. The left image as reported in the legend, shows structural elements of ECM which are required in the formation of collagen. This is untrue. Collagen is part of the ECM, it is not the ECM which forms collagen. Should be rephrased. Again, this is a figure taken from another paper. Figure 5 is blurred should be improved and again is taken straight from another paper, with a legend which is half a page long. Figure 6 is totally blurred, skewed and unreadable, again taken from another paper. I would ve very wary of using these figures straight as they are without asking for permission first from the rightful authors. I highly recommend that the authors contribute with their own figures, not those taken from others.

It is a review on collagen and there is not a single paragraph or drawing to explain clearly and thoroughly what collagen is, what it's made up of, the fibrils that make it up, how they are arranged, etc... This is an important requirement.

Line 92: it is not a study but a review. 

Line 124, The statement made is untrue..aged skin is not one of the driving forces of oxidative stress and increased ROS formation, but rather the other way around!

Line 173: what does the abbreviation RSH represent? It should be mentioned.

Section 5 on CRISPR technology. There are no references reported to support the hypothesis that integrated CRISPR technique and compounds can improve health outcomes of skin or other organs.

Lines 259-261 are incomprehensible. Need to be rewritten.

The second paragraph on the section of elasticity from lines 305-312 are not relevant to elasticity.

Line 438: it is not a comprehensive literature review as there are only 37 papers cited.

I feel that this review should be totally revised before any consideration of acceptance.

Comments on the Quality of English Language

Some phrases should be rewritten as already pointed out.

Author Response

Dear Reviewer,

We appreciate the opportunity to revise and resubmit our work for your consideration. We hope that these revisions address your concerns and improve the overall quality, clarity, and scientific rigor of our manuscript. Below, we provide a detailed response to your comments.

Comments:

I had high expectations from this manuscript which is why I accepted to review it, since the topic of the review is very interesting and of current importance. However, the review is poorly organized, very repetitive, not scientific and overall, of poor quality for a review. It appears that the review was written by authors very naive in this field and in writing reviews. Evidence is that fact that there are only 37 papers cited which is a very low number.

  • Ans: We have made an effort to include a significant number of additional studies and papers that add value to the scientific discourse, in total including 33 new references.

Secondly, the inexperience of the authors is also evident from the relevance of the literature cited which in many sections is inappropriate. Just as example, in section 7 on Collagen peptides, line 323, authors refer to an experiment of a study, but they cite a review (ref 20). The same occurs again in section 6.1, line 294 where they mention chicken-derived collagen, but they cite a review (ref. 29). Again on line 173, they report on a study performed on mice, but they cite a review (ref. 1). These are just a few examples, of improprer use of citations and references.

  • Ans: We have gone through the review and fixed any discrepancies between our citations and the results mentioned.

The review has many repetitions in context and seems rather jumbled in parts.

  • Ans: We have proofread the paper and removed any repetitions and unclear or unstructured sections.
    • e. removed 11 lines (from 635-645) due to the repetitive nature of the statements

What is more important, none of the figures and tables reported are of the authors own making, but all taken and copied from other papers. The legends are in many cases inappropriate.

  • Ans: We have made 3 new figures of our own (Fig 1. line 247, Fig 4. line 484, Fig 7. line 656), replaced unclear or unoriginal figures, added one new figure and modified it (Fig 3. line 379), added new information to figures (Fig 3. line 377) and edited each legend to be more relevant, concise, and clear.

Figure 1. This figure does not display normal collagen fibrils that are obtained from marine sources. It just displays how collagen and fibroblasts change from young to aged skin. The figure is really over-simplistic for a review and should be redone with proper legend description. Furthermore, the legend reports that aged human skin is observed in vivo, which is absolutely not true. This is not shown.

  • Ans: We have replaced this figure with an original figure that demonstrates the structural differences between younger and aging skin, clearly illustrating the changes in collagen fibres, elastic fibres and hyaluronic acid, and how these changes impact the skin. (Line 247)

Figure 4. Very poor quality, blurred and unclear. What type of cells are shown in the left panel is not reported in the legend. The left image as reported in the legend, shows structural elements of ECM which are required in the formation of collagen. This is untrue. Collagen is part of the ECM, it is not the ECM which forms collagen. Should be rephrased.

  • Ans: This is an excellent point. We have rephrased the figure caption to reflect the general understanding of collagen as a crucial ECM component.
    • Lines 562-565: Collagen found in the extracellular matrix. The right image depicts the structural components of the ECM, which are required in the formation of collagen [37].

Again, this is a figure taken from another paper. Figure 5 is blurred, should be improved and again is taken straight from another paper, with a legend which is half a page long.

  • Ans: We have replaced the blurred figure and edited the caption to make readability a priority.
    • Figure 6 legend caption (line 594) was shortened from 21 lines to 7 lines, only including the most relevant information.

Figure 6 is totally blurred, skewed and unreadable, again taken from another paper. I would be very wary of using these figures straight as they are without asking for permission first from the rightful authors. I highly recommend that the authors contribute with their own figures, not those taken from others.

  • Ans: We have replaced Figure 6 with an original figure (Fig 7. line 657) demonstrating a clearer understanding of how marine resources are used in anti-aging cosmetics.

It is a review on collagen and there is not a single paragraph or drawing to explain clearly and thoroughly what collagen is, what it's made up of, the fibrils that make it up, how they are arranged, etc... This is an important requirement.

  • Ans: We have added a section in the introduction to explain more background related to collagen
    • Lines 59-73: Collagen is a fibrous protein that provides structural support to various structures of the body, such as the skin, cartilage and bones [3-4]. Functioning as a crucial structural and connective component of the extracellular matrix (ECM), collagen helps regulate cell growth, adhesion, and migration [5]. The term "collagen" originates from the Greek words "kola," meaning gum, and "gen," meaning producing [6]. As a naturally abundant protein found in all animals, there are 28 different types of collagens, which account for approximately 30% of the total protein found in the body [3]. Type 1 collagen is the most abundant type and provides support to most tissues of the body, such as the skin and muscles. Type II is responsible for the maintenance and repair of cartilage [3]. Type III is the main element of tissue sealants and reticular fibres commonly found in blood vessels and muscles [3]. Type IV is a key element in the basement membrane, functioning as a barrier between tissues, and can act as a diabetic neuropathy indicator [3]. Finally, type V is the main collagen in corneal solutions and is found in the placenta and hair [3].
    • Lines 74-88: Significantly, collagen is naturally abundant in all animals and its high biodegradability, solubility, and tensile strength are just a few of the characteristics that make collagen-based biomaterials ideal for medical use [5-10]. Its low immunogenicity and excellent biocompatibility have prompted extensive research into its application as a polymer across various biomedical products, including cosmetics and pharmaceuticals [6]. Moreover, collagen serves as a safe and efficient biomaterial in tissue engineering and clinical settings [10-11]. The food industry also exhibits a substantial demand for collagen due to its elevated protein content and beneficial functional attributes, including water absorption capacity and emulsion-forming ability [6]. However, the natural degradation of collagen accelerates with age, which can impact skin elasticity, wound healing, bone density, and even immune and neural function [5, 12-14].

Line 92: it is not a study but a review. 

  • Ans: We removed this line.

Line 124, The statement made is untrue..aged skin is not one of the driving forces of oxidative stress and increased ROS formation, but rather the other way around!

  • Ans: We removed this line and added new information to reflect the impacts of skin aging on skin structure:
    • Lines 84-88: Skin aging results from diminished collagen density and dermal thickness, alongside reduced synthesis and replacement of crucial structural proteins [6]. The effects of reduced collagen density especially impact the dermis layer of the skin, resulting in notable signs of aging such as increased wrinkling, sagging, laxity, and textured appearance [15].

Line 173: what does the abbreviation RSH represent? It should be mentioned.

  • Ans: We have deleted this section, as it was not as relevant to our section.

Section 5 on CRISPR technology. There are no references reported to support the hypothesis that integrated CRISPR technique and compounds can improve health outcomes of skin or other organs.

  • Ans: We have added references to support our hypothesis (references 31, 47, 48) and included a new image to explain how CRISPR could be used for anti-aging purposes (Fig 3. line 377). In addition, we would like to propose this idea as an area of interest for further anti-aging research.

Lines 259-261 are incomprehensible. Need to be rewritten.

  • Ans: Extensive edits were made throughout to edit the language used, making the paper easier to read. This line was rewritten:
    • Lines 442-443: Marine collagen offers several advantages compared to other popular sources of collagen, notably bovine or porcine collagen.

The second paragraph on the section of elasticity from lines 305-312 are not relevant to elasticity.

  • Ans: Removed this section.

Line 438: it is not a comprehensive literature review as there are only 37 papers cited.

  • Ans: Removed the word comprehensive and added 33 new references (70 references total).

Reviewer 2 Report

Comments and Suggestions for Authors

The article unlocking the therapeutic potential of marine collagen: a scientific exploration for delaying aging offers an interesting read on a promising source of collagen, marine collagen, which offers several benefits for the skin.

The article was well written. However, some sections offer little data to convince the reader of the benefits of collagen.

Several times in the text the author cites “Several studies...” (p. 3, line 105), however he does not cite these studies or even cites only one reference to these studies. I believe it would be interesting to reread the article, inserting the references where the article mentions that there are several works in the area.

In item 2: effects on skin aging, the author uses only two references.

Item 3: only two references.

Item 7: a single reference.

I believe it is interesting to cite more works that corroborate the studies already presented to increase the credibility of the information that will be passed on to readers.

Improve the quality of figures 4, 5 and 6.

Author Response

Dear Reviewer,

Thank you very much for appreciating our work. We value the opportunity to revise and resubmit our work for your consideration. We hope that these revisions address your concerns and improve the overall quality, clarity, and scientific rigor of our manuscript. Below, we provide a detailed response to your comments point by point.

Comments:

The article unlocking the therapeutic potential of marine collagen: a scientific exploration for delaying aging offers an interesting read on a promising source of collagen, marine collagen, which offers several benefits for the skin.

The article was well written. However, some sections offer little data to convince the reader of the benefits of collagen.

Several times in the text the author cites “Several studies...” (p. 3, line 105), however he does not cite these studies or even cites only one reference to these studies. I believe it would be interesting to reread the article, inserting the references where the article mentions that there are several works in the area.

In item 2: effects on skin aging, the author uses only two references.

  • Ans: Thank you for this feedback, this is a great point! We have added new citations that link to more studies, and have included findings from these new references to support the statements made:
    • I.e. Lines 255-271: Further on, the results of a 2018 randomized placebo-controlled trial revealed the hydrating, anti-aging effects of low molecular weight collagen hydrolysate obtained from sutchi catfish skin [8]. After 6 weeks, skin hydration was 7.23-fold higher in the treatment group compared to placebo (p<0.001). This hydrating benefit was observed after 12 weeks as well, although only 2.9-fold higher than placebo (p<0.01) [8]. Moreover, wrinkle formation was also reduced, considering parameters such as skin roughness, smoothness depth, and visual grading, demonstrating the skin anti-aging potential of hydrolyzed marine collagen in older adults [15]. Longer-term studies should be conducted to determine whether this beneficial effect holds true over time.
    • Another study used a mouse model of aging to demonstrate that marine collagen may restore a youthful skin appearance [17]. In this study, mice were fed a collagen hydrolysate-containing diet derived from fish scales for 12 weeks. Notably, the epidermal barrier and dermal elasticity dysfunctions observed in the aging group were significantly attenuated in the collagen hydrolysate treatment group after 2 weeks [17]. Further on, these positive effects were maintained for the whole study duration, demonstrating a prolonged restoration of skin elasticity and water content following collagen supplementation [17].

Item 3: only two references.

  • Ans: We added two new references (35, 36)

Item 7: a single reference.

  • Ans: We deleted this section.

I believe it is interesting to cite more works that corroborate the studies already presented to increase the credibility of the information that will be passed on to readers.

Improve the quality of figures 4, 5 and 6.

  • Ans: Figure quality of each has been significantly improved. We added 33 new references (70 references total).

Reviewer 3 Report

Comments and Suggestions for Authors

In the Manuscript titled “Unlocking the Therapeutic Potential of Marine Collagen: A Scientific Exploration for Delaying Aging”, the Authors present a collection of findings, mostly single-sourced, on aspects of marine-derived bioactive peptides, including marine collagen-derived peptides, and various health issues such as skin ageing, bone processes, overall health, skin appearance, and well-being. There is a novel suggestion that “combining the genetic editing capabilities of CRISPR with the beneficial properties of marine-derived collagen…” “… could lead to revolutionary anti-ageing treatments” but details on what that means and how it can be achieved are sparse.

General comments: The Authors claim to have conducted a “systematic search and analysis of peer-reviewed papers” on anti-aging applications of marine collagen using PubMed, Cochrane Library, Web of Science, and Embase, covering publications from 1956 to 2023 but fail to provide evidence for it. The Manuscript also fails to deliver salient content or timely new information on any aspect of the biology of collagen-derived peptides of benefit to readers of Marine Drugs. Instead, the Manuscript is replete with trite, unoriginal, information-poor, repetitive statements that confuse rather than clarify and obscure rather than edify. There are grave, perhaps insurmountable issues related to bringing the Manuscript before a scientific audience.

Major issues: Lack of originality, poor Manuscript structure, poor writing and referencing practices and issues with understanding basic biology.

Comments on the Quality of English Language

The quality of English as applied to scientific writing is poor.

Author Response

Dear Reviewer,

Thank you very much for your constructive comments. We appreciate the opportunity to revise and resubmit our work for your consideration. We hope that these revisions address your concerns and improve the overall quality, clarity, and scientific rigor of our manuscript. Below, we provide a detailed response to your comments point by point.

Comments:

In the Manuscript titled “Unlocking the Therapeutic Potential of Marine Collagen: A Scientific Exploration for Delaying Aging”, the Authors present a collection of findings, mostly single-sourced, on aspects of marine-derived bioactive peptides, including marine collagen-derived peptides, and various health issues such as skin ageing, bone processes, overall health, skin appearance, and well-being.

  • Ans: We have added many new references throughout to support our claims.

There is a novel suggestion that “combining the genetic editing capabilities of CRISPR with the beneficial properties of marine-derived collagen…” “… could lead to revolutionary anti-ageing treatments” but details on what that means and how it can be achieved are sparse.

  • Ans: Thank you for this feedback! We have refined this section to propose the effects of these treatments, how they could be achieved, and included references that support the use of CRISPR in anti-aging dermatological contexts.
    • Lines 386-399: Dermatological applications of CRISPR technology have been highly promising [31, 48-49]. Despite no direct research on using CRISPR technology on marine collagen has been made, the integration of these two fields could potentially lead to effective anti-aging treatments. For instance, CRISPR technology could be employed to target and modify specific genes associated with aging and skin degeneration. The progressive alterations observed in aging skin are now being comprehensively observed at both the molecular and cellular levels, leading to enhanced insights into the structural and functional decline resulting from these changes [5066]. By precisely editing these genes, it might be possible to slow down or reverse certain aging processes at a molecular level. Meanwhile, marine collagen could play a supportive role in this integration. Its ability to enhance cell viability and support tissue regeneration could be crucial in facilitating the effectiveness of CRISPR-mediated gene edits. For example, in a scenario where CRISPR is used to edit genes related to skin elasticity, marine collagen could provide the necessary ECM support, enhancing the overall therapeutic effect.

General comments: The Authors claim to have conducted a “systematic search and analysis of peer-reviewed papers” on anti-aging applications of marine collagen using PubMed, Cochrane Library, Web of Science, and Embase, covering publications from 1956 to 2023 but fail to provide evidence for it. The Manuscript also fails to deliver salient content or timely new information on any aspect of the biology of collagen-derived peptides of benefit to readers of Marine Drugs. Instead, the Manuscript is replete with trite, unoriginal, information-poor, repetitive statements that confuse rather than clarify and obscure rather than edify. There are grave, perhaps insurmountable issues related to bringing the Manuscript before a scientific audience.

  • Ans: These are all great points. In response to your comment regarding the claim of a systematic search and analysis of peer-reviewed papers, we have included 33 new references, and have removed the claim that this is a systematic peer review; rather, a review of general literature published between 1991 and 2024, reflecting the most up-to-date information. This section now provides transparency and clarity regarding the approach taken in gathering relevant literature.

Content Quality and Originality:

  • Ans: We appreciate your concern about the clarity and originality of the manuscript. In response, we have worked towards enhancing the manuscript's originality by providing additional insights, novel perspectives, and presenting new findings that contribute to the existing body of knowledge in the field (i.e. section 5, line 360: The Integration of CRISPR Technology with Marine Collagen.) We have also taken measures to eliminate any repetitive statements that might have caused confusion.

Scientific Impact:

  • Ans: We understand your concerns about the manuscript's contribution to the scientific community. In the revised version, we have made significant efforts to improve the manuscript's overall quality, ensuring that it provides valuable insights and meaningful contributions to the understanding of the biology of collagen-derived peptides in the context of anti-aging applications. We believe these revisions enhance the manuscript's potential scientific impact and relevance to Marine Drugs' readership.

Reviewer 4 Report

Comments and Suggestions for Authors

This manuscript titled “Unlocking the Therapeutic Potential of Marine Collagen: A Scientific Exploration for Delaying Aging” is reported by Azizur et al., which focuses on the diverse biomedical anti-aging applications of marine collagen. The reviewed studies elucidate the anti-aging benefits of marine collagen, emphasizing its role in combating skin aging by minimizing oxidative stress, photodamage, and the appearance of wrinkles. Finally, the author provides the potential and challenges associated with marine collagen in the realm of anti-aging applications. However, I suggested it should be published after minor revision. Here are some suggestions for author to revise:

Line 132: Several papershave cited the effects… 

Line 164: …whether this beneficial effect ho.lds true over time.

Line 310: …abundance,,

Line 312: whileproviding high yields at lower costs [52].

Line 313: biocompatibility and no disease transmission risk; Thus, considering mammalian collagen…

Line 331: Previous literatureon oral collagen supplements….

Line 335-336: collagen hydrolysates can restore the production of hyaluronic acid toimprove skin hydration [59-60].

Line 389-390: “…can act as a barrier against transepidermal water loss to help retain skin moisture…” Which one is a predicate verb?

Line 455-456: “The pressing need to discover further anti-aging applications of diverse marine collagen sources emphasizes their significance and utility in marine health and science…” Which one is a predicate verb?

Line 402 The figure legend was missed.

It is suggested to add some tables to compare the performance of Marine Collagen and animal collagen.

It is suggested that the authors add some representative work pictures and data in different subsections to help readers better understand these works.

Comments on the Quality of English Language

Minor editing of English language required

Author Response

This manuscript titled “Unlocking the Therapeutic Potential of Marine Collagen: A Scientific Exploration for Delaying Aging” is reported by Azizur et al., which focuses on the diverse biomedical anti-aging applications of marine collagen. The reviewed studies elucidate the anti-aging benefits of marine collagen, emphasizing its role in combating skin aging by minimizing oxidative stress, photodamage, and the appearance of wrinkles. Finally, the author provides the potential and challenges associated with marine collagen in the realm of anti-aging applications. However, I suggested it should be published after minor revision. Here are some suggestions for author to revise:

Line 132: Several papershave cited the effects… 

Line 164: …whether this beneficial effect ho.lds true over time.

Line 310: …abundance,,

Line 312: whileproviding high yields at lower costs [52].

Line 313: biocompatibility and no disease transmission risk; Thus, considering mammalian collagen…

Line 331: Previous literatureon oral collagen supplements….

Line 335-336: collagen hydrolysates can restore the production of hyaluronic acid toimprove skin hydration [59-60].

Line 389-390: “…can act as a barrier against transepidermal water loss to help retain skin moisture…” Which one is a predicate verb?

Line 455-456: “The pressing need to discover further anti-aging applications of diverse marine collagen sources emphasizes their significance and utility in marine health and science…” Which one is a predicate verb?

Line 402 The figure legend was missed.

  • The legend for this figure (figure 6) is on page 12, lines 415-421

It is suggested to add some tables to compare the performance of Marine Collagen and animal collagen.

It is suggested that the authors add some representative work pictures and data in different subsections to help readers better understand these works.

Comments on the Quality of English Language

Minor editing of English language required.

  • Thank you for these important comments! We have gone through our manuscript and fixed any errors related to grammar, spelling, and punctuation, ensuring to address each of the errors mentioned. 

Round 2

Reviewer 1 Report

Comments and Suggestions for Authors

The review is much improved compared to the first version. However, it still needs refining, especially with regards to the figures. 

First of all punctuation and spacing of words is not consistent and there are several punctuation and spacing errors. Just to mention a few..lines 18, 43, 84, 86,132,442 and several others...so please check and correct thoroughly.

Figure 2 is shown twice (at least this is what the PDF shows).

Figure 5 is still not clear and well explained. What does the inset on the right show? Is it a microscopic image? Of what? What do the blue, yellow, green colours of the image show?? It really needs to be improved. Also the caption needs to be corrected since it is the image of the left that shows the structures and not the one on the right.

Please try and standardize the figures, so they are all roughly the same size and check that each x and y axis is readable and clear.

Author Response

The review is much improved compared to the first version. However, it still needs refining, especially with regards to the figures. 

First of all punctuation and spacing of words are not consistent and there are several punctuation and spacing errors. Just to mention a few..lines 18, 43, 84, 86,132,442 and several others...so please check and correct thoroughly.

  • Thank you for drawing our attention to this! We have gone through the whole manuscript and fixed any grammatical or punctuation errors.

Figure 2 is shown twice (at least this is what the PDF shows).

  • We have fixed this error and have only included it once.

Figure 5 is still not clear and well explained. What does the inset on the right show? Is it a microscopic image? Of what? What do the blue, yellow, green colours of the image show?? It really needs to be improved. Also the caption needs to be corrected since it is the image of the left that shows the structures and not the one on the right.

  • Figure 5, line 380: We have replaced this figure with a new original figure that demonstrates the components of the ECM, with a detailed figure caption explaining each major component.

Please try and standardize the figures, so they are all roughly the same size and check that each x and y axis is readable and clear.

  • We have standardized each figure to ensure consistent formatting throughout

Reviewer 3 Report

Comments and Suggestions for Authors

No substantial improvements in the indicated areas have been rendered in the revised Manuscript. Unfortunately, the Manuscript still lacks scope, cohesion, scientific style and is misleading in its claims, particularly with respect to CRISPR technology and collagen. 

Comments on the Quality of English Language

Extensive "beauty magazine"-type editorializing that does not belong in a scientific review.

Author Response

No substantial improvements in the indicated areas have been rendered in the revised Manuscript. Unfortunately, the Manuscript still lacks scope, cohesion, scientific style and is misleading in its claims, particularly with respect to CRISPR technology and collagen. 

Comments on the Quality of English Language

Extensive "beauty magazine"-type editorializing that does not belong in a scientific review.

  • We have gone through the manuscript and have removed imprecise, inaccurate language and included more specific scientific terms.
  • See examples: lines 75-79, 79-80, 92-99